# Utilization of maternal waiting home and associated factors among women who gave birth in the last one year, Dabat district, Northwest Ethiopia

**Mulugeta Melese Shiferaw[1], Agumas Eskezia Tiguh**[2]*****, **Azmeraw Ambachew Kebede[2], Birhan Tsegaw Taye**[3]

1 Coordinator of Dabat Health Center, Gondar, Ethiopia, 2 Department of Clinical Midwifery, School of Midwifery, College of Medicine and Health science, University of Gondar, Gondar, Ethiopia, 3 Department of Midwifery, College of Health science, Debre Birhan University, Debre Birhan, Ethiopia

* aeskezia@gmail.com

**Data Availability Statement:** All relevant data are within the manuscript and its Supporting Information files.

## Abstract

### Background

Maternal mortality and adverse pregnancy outcomes are still challenges in developing countries. In Ethiopia, long distances and lack of transportation are the main geographic barriers for pregnant women to utilize a skilled birth attendant. To alleviate this problem, maternity waiting homes are a gateway for women to deliver at the health facilities, thereby helping towards the reduction of the alarming maternal mortality trend and negative pregnancy outcomes. However, there is a paucity of evidence regarding the utilization of maternity waiting homes in the study area. Therefore, this study aimed to assess utilization of maternity waiting home services and associated factors among mothers who gave birth in the last year in Dabat district, northwest Ethiopia.

### Methods

A community-based cross-sectional study was conducted from January 5 to February 30, 2019. A total of 402 eligible women were selected using a simple random sampling technique. Data were collected using a structured, pre-tested, and interviewer-administered questionnaire through face-to-face interviews. Data were entered into EPI info version 7.1.2 and exported to SPSS version 20 for analysis. Both bivariable and multivariable logistic regression models were fitted. Statistically significant associations between variables were determined based on the adjusted odds ratio (AOR) with its 95% confidence interval and p-value of $\leq 0.05$.

### Results

Maternity waiting home utilization by pregnant women was found to be 16.2% (95% CI: 13, 20). The mothers' age (26–30 years) (AOR = 0.24; 95% CI: 0.08,0.69), primary level of education (AOR = 9.05; 95% CI: 3.83, 21.43), accepted length of stay in maternity waiting

**Funding:** This study was sponsored by the University of Gondar. However, the funder had no role in study design, data collection and analysis, decision to publish, or preparation of the manuscript.

**Competing interests:** The authors have declared that no competing interests exist.

**Abbreviations:** ANC, Antenatal care; EDHS, Ethiopian demographic and health survey; MMR, Maternal mortality ratio; MWH, Maternal waiting home; MWHS, Maternal waiting home services; RHSP, Reproductive health strategic plane; WHO, World Health Organization.

homes (AOR = 3.15; 95% CI: 1.54, 6.43), adequate knowledge of pregnancy danger signs (AOR = 7.88; 95% CI: 3.72,16.69), jointly decision on the mother's health (AOR = 2.76; 95% CI: 1.08,7.05), and getting people for household activities (AOR = 2.59, 95% CI: 1.21, 5.52) had significant association with maternity waiting home utilization.

## Conclusion

In this study, maternity waiting home utilization was low. Thus, expanding a strategy to improve women's educational status, health education communication regarding danger signs of pregnancy, empowering women's decision-making power, and shortening the length of stay at maternity waiting homes may enhance maternity waiting home utilization.

## Introduction

Maternity waiting homes (MWH) are temporary residences in which high-risk pregnant women or women residing far from healthcare facilities can wait in their last few weeks of pregnancy before giving birth. It is an effective strategy to promote safe delivery by a skilled health provider and help rapidly access emergency obstetric care when a complication arises [1].

In sub-Saharan Africa (SSA), the majority of births have been attended without a skilled healthcare provider. As a result, in 2017, there were about 196, 000 maternal deaths in the region, which accounts for 66% of the global maternal death rate [2]. Ethiopia is also one of the SSA countries with low maternal health service utilization and a high maternal mortality ratio. Thus, the facility's birth rates are as low as 28%, and about 1400 maternal deaths occur annually [2].

Although there has been a remarkable decrement in maternal mortality ratio (from 871 in 2000 to 401 per 100, 000 live births in 2019), Ethiopia is still among the top countries having the highest maternal death record globally [3]. Furthermore, maternal health service coverage has increased only slightly, with antenatal care (ANC) and skilled birth attendant coverage increasing by 62% and 26%, respectively, in 2016 [4]. However, there is a mismatch between ANC and skilled birth attendance coverage. This suggests that women receiving ANC were giving birth at home for a variety of reasons, including long distances to reach a health facility, delayed transportation, and/or family influences [5–8]. According to available evidence, 70% of health centers in Ethiopia and 73% in the Amhara region had MWHs [9], which is close to the Ethiopian reproductive health strategic plan (RHSP). The RHSP was targeted to equip 75% of health centers with MWH by 2020 [10]. However, the availability and utilization of MWHS are limited, in that, only 44% of women utilize MWHS in Ethiopia [9].

Even though the risk of pregnancy varies from one mother to another, any woman may develop unexpected complications during pregnancy and childbirth [11]. So, for timely and appropriate intervention, expanding MWHS utilization is a precious strategy. As a result, the establishment of MWH at each health center is strongly recommended and supported by the World Health Organization (WHO), particularly in developing countries [12]. Studies have shown that MWHS have a significant role in reducing maternal and perinatal mortalities [8, 13]. It has been evidenced that utilization of MWHS reduces maternal mortality by 80% and stillbirth rates by 73% in developing countries [8]. Another study in Ethiopia found that hospitals with MWHS reduced perinatal mortality by 47% and direct obstetric complications by 49% [14]. In addition, it urges women to use maternal healthcare services like skilled birth

attendants and other comprehensive emergency obstetrics care, thereby reducing negative pregnancy outcomes [13, 15, 16]. Moreover, MWHS can decrease the gap between urban-rural maternal health service utilization [5].

In Ethiopia, there are limited governmental reports and published studies on actual MWHS utilization. Some of the published studies focus merely on the physical establishment of MWH at health institutions rather than utilization [9, 14]. Some other studies focus on the intention to use rather than the actual utilization of MWHS [17–21]. Although some other studies conducted on the utilization of MWHS, they collected the data at health institutions [22], and failed to address the reasons for the non-utilization of MWHS for home-delivered women. As a result, this study will add further important information on the utilization of MWH to be revised timely at every level to fill the possible gaps. Therefore, the current study was aimed to assess the utilization of MWHS and associated factors in northwest Ethiopia.

## Method and materials

### Study design, setting, and period

A community-based cross-sectional study was conducted from January 5 to February 30, 2019. The study was conducted in Dabat district, Amhara regional state, northwest Ethiopia, which is located about 245 km northwest of Bahir Dar (the capital city of Amhara regional state), and 70 km away from Gondar city. Dabat district has six administrative sub-divisions. Besides, there are a total of six health centers (one in each subdivision) in which only the four sub-divisions have maternity waiting homes.

### Study population

All women who gave birth in the last year before the study period in the selected clusters were the study population. Women who were seriously ill throughout the data collection period were excluded.

### Sample size determination and sampling procedure

The sample size for this study was determined by using a single proportion formula by considering the following assumptions; the prevalence of MWHs, 38.7%[18], 95% level of confidence, and a 5% margin of error. Therefore, $\frac{(Z\alpha/2)^2 p(1-p)}{d^2} = \frac{(1.96)^2 * 0.387(1-0.387)}{(0.05)2} = 365$. Where, n = required sample sizes, $\alpha$ = level of significance, z = standard normal distribution curve value for 95% confidence level = 1.96, p = proportion of maternity waiting home utilization, and d = margin of error. By considering a 10% non-response rate, the final minimum adequate sample size was 402. Dabat district has 6 administrative subdivisions, of these, only four of the subdivisions have MWH. A survey was conducted in the four subdivisions of the district with the assistance of health extension workers to identify women who were eligible for the study. Following the identification of the study population, a sampling frame was designed by compiling the list of all women in the four districts. Proportional allocation was done to each of the four subdivisions to draw the final sample size. Lastly, the study subjects were selected by using a simple random sampling technique (Fig 1).

### Study variables

**Dependent variable.**  Utilization of MWHS (utilized/ not utilized)

**Independent variables.**  Socio-demographic characteristics; Age of the mother, religion, marital status, occupation, educational status, partner's educational status, time taken to reach

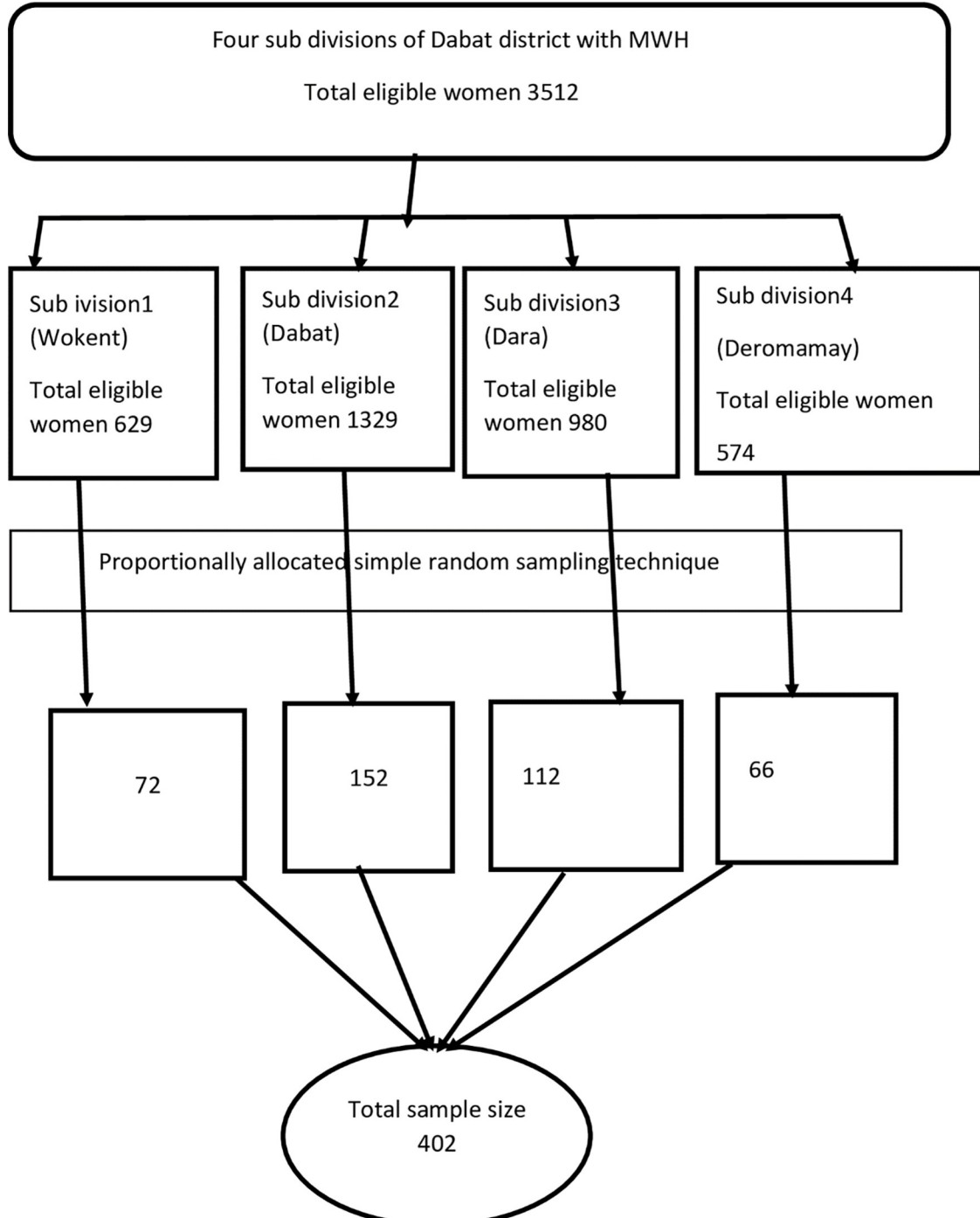

**Fig 1. Schematic presentation of the sampling procedure among women who gave birth in the last one year prior to the study period in Dabat District, North West, Ethiopia, 2019.**

health facilities, transportation access to the health facilities, affordability of transport cost, way of transportation.

Reproductive health and obstetrics related; Decision power of mother on own health, number of live birth, history of stillbirth, the birthplace of the last child, number of ANC visits of

the last pregnancy, planned or unplanned pregnancy, place of ANC visit, information on birth preparedness plan, knowledge of danger sign during pregnancy, and awareness of expected date of delivery.

Social and behavioral factors: possibility of getting people for household activities, getting people for a chilled caregiver, perceived the two-four weeks' duration stay before labor at MWH is acceptable, the possibility of being away from the work.

## Operational definitions

**Maternity waiting home utilization.**   Those women who stayed in the MWH before delivery starting from 24 weeks of pregnancy duration and above in the last pregnancy [1].

**Knowledge on danger sign of pregnancy.**   A woman who list out three and more danger signs of pregnancy (Vaginal bleeding, gush of fluid per vagina, severe abdominal pain, high grade fever, fainting, decreased fetal movement, blurred vision, severe headache, edema or body swelling) was considered as knowledgeable [23–25].

**Accepted length of stay.**   Women's perception of the length of two-four weeks is optimal.

## Data collection tools and procedures

Data were collected using a pre-tested, semi-structured, and interviewer-administered questionnaire through face-to-face interviews. The study tool was prepared by reviewing related literature [18, 23, 26]. The questionnaire was first developed in English and then translated into the Amharic language, and then back to English to keep its consistency. Four diploma and one BSc midwives were employed for data collection and supervision, respectively.

## Data quality assurance

Before the actual date collection period, a pretest was done on 5% of the calculated sample size outside of the study area. Data collectors were trained on data collection techniques for one day. Supervision was followed regularly during the data collection period, and the collected data were checked daily for completeness and consistency.

## Data processing and analysis

Data cleaning was performed to check for accuracy, completeness, consistencies, and missing values. After the data had been checked for completeness and accuracy, it was coded manually and then entered into Epi-Info version 7.1.2 and exported to SPSS version 20 for analysis. Descriptive data were presented by tables, graphs, charts, frequencies, and proportions. Binary logistic regression was used to identify statistically significant independent variables, and variables having a p-value of $\leq 0.25$ in the bivariable logistic regression analysis were included in the multivariable logistic regression analysis to adjust for possible confounding factors. The adjusted odds ratio with a 95% confidence interval was used to determine the degree and direction of association between covariates and the outcome variable. The level of significance in the last model was declared at a p-value of $\leq 0.05$.

## Ethical consideration

Ethical clearance was obtained from the school of midwifery Ethical Review Committee under the delegation of the Institutional Review Board (IRB) of the University of Gondar with reference number (SMIDW/19/498/2018). A formal letter of approval was taken from Dabat district administrative health office. The purpose of the study was explained to the study participants, and written informed consent was obtained from every study participant before

data collection. For participants aged <18, written informed assent was taken from their parents.

## Result

### Socio-demographic characteristics of the study participants

A total of 402 women were participated in this study, with a response rate of 100%. The mean age of the study participants was 29.58 years (SD ±7.9) and 110 (27.4%) of them were in the age group of 26–30 years. The majority, 343 (85%) of the study participants and 208 (51%) of their husbands have no formal education. About 95 (23.6%) mothers traveled for more than two hours to reach the nearest health facility (Table 1).

**Table 1. Sociodemographic characteristics of the study participants in Dabat district northwest Ethiopia, 2019.**

| Variable | Number | Percent |
|---|---|---|
| **Age** | | |
| ≤20 | 14 | 3.5 |
| 21–25 | 103 | 25.6 |
| 26–30 | 110 | 27.4 |
| 31–35 | 90 | 22.4 |
| ≥36 | 85 | 21.1 |
| **Marital status** | | |
| Unmarried | 30 | 7.5 |
| Married | 372 | 92.5 |
| level of education | | |
| No formal education | 343 | 85.3 |
| Primary education | 47 | 11.7 |
| Secondary and above | 12 | 3 |
| **Mothers occupation** | | |
| Marchant | 132 | 32.84 |
| Farmer | 249 | 61.94 |
| Employee | 21 | 5.22 |
| **Husband/father of child/ Level of education** | | |
| No formal education | 208 | 51.74 |
| Primary education | 124 | 30.85 |
| Secondary and above | 70 | 17.41 |
| **Time taken to the nearest Health center** | | |
| ≤ 30 min | 82 | 20.40 |
| 31-60min | 84 | 20.90 |
| 61–90 min | 91 | 22.63 |
| 91-120min | 50 | 12.44 |
| ≥121 min | 95 | 23.63 |
| **Accessibility of transportation** | | |
| Easy to gate | 67 | 16.7 |
| Hard to gate | 335 | 83.3 |
| **Affordability of transport cost** | | |
| Affordable | 307 | 76.4 |
| Not affordable | 95 | 23.6 |
| **Way of transport if childbirth complications happen** | | |
| By car or Ambulance | 182 | 45.3 |
| Traditional means of transport | 220 | 54.7 |

## Reproductive and obstetrics related variables

One hundred ninety-one (47.5%) of the study participants had a joint decision with their husbands on their health. More than one-third, 148 (36.8%) of the study participants had three to four children, and 102 (25.4%) of the study participants had a history of stillbirth. Three-hundred fifty (81.7%) study participants had two or more ANC visits in their most recent pregnancy, and 167 (41.5%) women gave their last birth at home. Two-hundred fifty-eight (64.2%) of the study participants had planned pregnancies. However, only 145 (35.8%) of the study participants know about the danger signs of pregnancy (Table 2).

## Social and behavioral characteristics

One hundred forty-nine (37.1%) study participants perceived the specified waiting time at MWHS as an acceptable time. About 71.9%, 71.6%, and 62.5% of the study participants could

**Table 2. Reproductive and obstetric related factors of the study participants in Dabat district, north west Ethiopia, 2019.**

| Variable | Number | Percent |
|---|---|---|
| **Decision on maternal health** | | |
| Mother | 89 | 22.1 |
| Husband | 122 | 30.3 |
| Jointly | 191 | 47.5 |
| **Total live birth** | | |
| ≤2 | 129 | 32.1 |
| 3–4 | 148 | 36.8 |
| ≥5 | 125 | 31.1 |
| **Total no of pregnancy** | | |
| 1–3 | 187 | 46.5 |
| 4–5 | 112 | 27.9 |
| ≥6 | 103 | 25.6 |
| **History of IUFD/Stillbirth** | | |
| NO | 300 | 74.6 |
| YES | 102 | 25.4 |
| **Number of ANC visit** | | |
| One or no ANC visit | 52 | 12.9 |
| Two and more ANC Visit | 350 | 87.1 |
| **Awareness on EDD** | | |
| No | 128 | 31.8 |
| Yes | 274 | 68.2 |
| **Last pregnancy status** | | |
| Planed | 258 | 64.2 |
| Not planed | 144 | 35.8 |
| **Information on Berth preparedness plan** | | |
| Yes | 316 | 21.4 |
| No | 86 | 78.6 |
| **Knowledge on Danger signs of pregnancy** | | |
| Not knowledgeable | 257 | 63.9 |
| Knowledgeable | 145 | 35.8 |
| **Birth place for the current child** | | |
| Health institution | 235 | 58.5 |
| Home | 167 | 41.5 |

**Table 3. Social and behavioral factors for MWHs utilization, Dabat district, northwest Ethiopia, 2019.**

| Variable | Number | Percent |
|---|---|---|
| **Acceptability of two-four weeks waiting time** | | |
| Acceptable | 149 | 37.1 |
| Not acceptable | 253 | 62.9 |
| **Possibility of getting people for house holed activities** | | |
| Possible | 113 | 28.1 |
| Not possible | 289 | 71.9 |
| **Possibility of getting people for a child caregiver** | | |
| Possible | 114 | 28.4 |
| Not possible | 288 | 71.6 |
| **Possibility of getting attendants at MWHS** | | |
| Possible | 140 | 34.8 |
| Not possible | 262 | 65.2 |

not easily get any person for the household activities, child caregiver, and attendant at MWHS, respectively (Table 3).

## Information on maternity waiting home service

Three hundred eleven (77.4%) study participants have information on the maternity waiting home service. However, more than one-fourth (26.6%) of the study participants didn't know the location of maternity waiting homes (Fig 2).

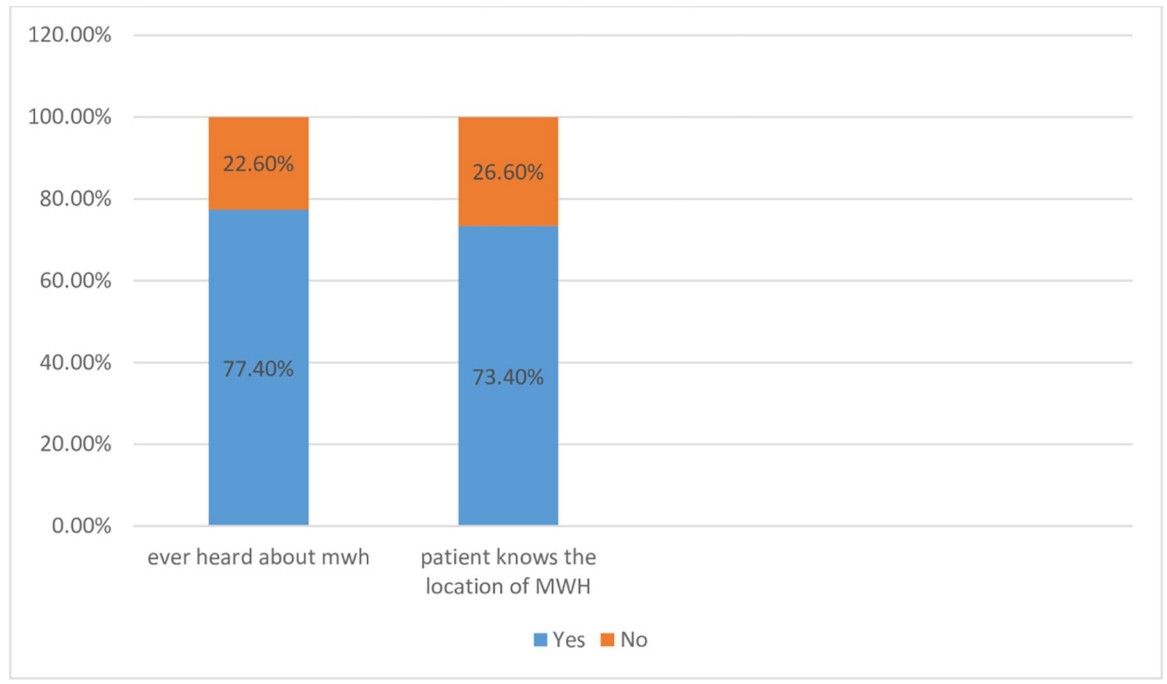

**Fig 2. Information about Utilization of maternity waiting home among women who gave birth in the last on year prior to the study period in Dabat District, north west Ethiopia, 2019.**

### Maternal waiting home utilization

Of the total study participants, only 16.2% (95% CI: 13, 20) of women used MWHS during their most recent pregnancy

### Factors associated with maternal waiting home utilization

Bivariable and multivariable logistic regressions were fitted to identify factors associated with MWHS utilization. From the multivariable logistic regression analysis, maternal age, level of education, maternal knowledge on dangers signs of pregnancy, decision on mother's health, the possibility of getting people for household care, the possibility of getting people for child care and acceptability of waiting time at MWH had an association with the utilization of MWHS. Those mothers whose age category was aged between 26–30 years old were 76% less likely to utilize MWHS than those women whose age category was 36 and above (AOR: 0.24; 95% CI: 0.087, 0.69). Mothers attending a primary level of education were 9.05 times more likely to utilize MWHS as compared to mothers who had no formal education (AOR: 9.05, 95% CI: 3.83, 21.43). Similarly, mothers having adequate knowledge of pregnancy danger signs were 7.88 times more likely to utilize MWHS than those women having less knowledge of danger signs of pregnancy (AOR: 7.88, 95% CI: 3.72,16.69). Likewise, mothers who had a shared decision-making power on their health status with their husbands were 2.76 times more likely to utilize MWHs than those who decide on their health condition by themselves (AOR: 2.76, 95% CI:1.08, 7.05). The odds of utilizing MWHS were 2.56 times higher among mothers who had people cover the household activities compared with their counterparts (AOR: 2.59, 95%CI: 1.21, 5.52). Lastly, mothers who accepted the specified duration of waiting time were 3.15 times more likely to utilize MWHs than those mothers who did not accept the specified waiting time duration (AOR: 3.15, 95% CI: 1.54, 6.43) (Table 4).

## Discussion

The public health importance of this study is to provide information for health managers, health care providers, and concerned stakeholders, and to identify the factors affecting MWHS utilization. Utilizing MWHS will create an opportunity for facility-based delivery, thereby decreasing maternal and perinatal morbidity and mortality. Therefore, this study was conducted to assess the utilization of MWHS and associated factors among mothers who gave birth in the last year in Dabat district, northwest Ethiopia.

Accordingly, the study revealed that MWH service utilization was 16.2%. This finding is lower than studies conducted in Ethiopia, including Jimma district (38.7%) (18), Bench Maji zone (39%) [27], and Arsi zone(23.6%) [22]. This finding is also lower compared to two studies conducted in Zambia, in which (27.3%) and (31%) of women used MWHS [16, 28]. The discrepancies might be due to the differences in study settings, in which the aforementioned studies were conducted at a facility level, whereas the current study was community-based. For instance, the study in the Jimma district included all women who gave birth in health facilities, so the chances of obtaining MWH- users might be high among those women. But in the current study, 41.5% of the participants gave birth at home for their most recent pregnancy, and no MWH-users were identified among women who gave birth at home. It is believed that admission into the MWH increases the chances of women giving birth at health facilities.

In contrast, the MWHS utilization of the current finding is higher than in another study conducted in the Jimma Zone, Ethiopia (7%) [26]. The possible discrepancy might be due to differences in the population background, wherein among the total study participants in the Jimma zone, only (30%) of them lived remotely from the health facilities (the distance from homes to the nearest health center takes 30 minutes and more). However, in this study, 67.6% of the participants traveled for more than an hour to reach the nearby health facilities.

**Table 4. Factor associated with maternal waiting home utilization, Dabat district, north west Ethiopia, 2019.**

| Variables | MWH utilize | MWH non utilize | COR(95%CI) | AOR(95%CI) | p-value |
|---|---|---|---|---|---|
| **Age** | | | | | |
| ≤20 | 5 | 9 | 1.59(.48,5.26) | 2.22(0.47,10.4) | 0.313 |
| 21–25 | 19 | 84 | .648(.32,1.29) | 0 .55(0.159,1.1) | 0.200 |
| 26–30 | 11 | 99 | .318(.44,.70) | **0 .24(0.087,0.69)** | **0.008** |
| 31–35 | 8 | 82 | .279(.117,.669) | 0 .36(0.13,1.02) | 0.057 |
| ≥36 | 22 | 63 | 1 | 1 | 0.021 |
| **level of education** | | | | | |
| No formal education | 38 | 305 | 1 | 1 | 0.000 |
| Primary level of education | 25 | 22 | 9.12(4.7,17.73) | **9.05(3.83,21.43)** | **0.000** |
| Secondary and above level of education | 2 | 10 | 1.6(.34,7.6) | 4.86(0.86,27.59) | 0.074 |
| **Husband's Level of education** | | | | | |
| No formal education | 29 | 179 | 1 | 1 | 0.569 |
| Primary level of education | 19 | 105 | 1.117(.59,2.1) | 1.21(.52,2.81) | 0.657 |
| Secondary and above level of education | 17 | 53 | 1.98(1.01,3.87) | 1.76(0.62,5.02) | 0.289 |
| **History of stile birth or IUFD** | | | | | |
| No | 42 | 258 | 1 | 1 | |
| Yes | 23 | 79 | 1.78(1.01,3.15) | 1.24(0.58,2.67) | 0.577 |
| **Pregnancy status** | | | | | |
| Un Planed | 36 | 222 | 1 | | |
| planed | 29 | 115 | 1.55(.908,2.66) | 1.19(0.59,2.44) | 0.619 |
| **No of ANC** | | | | | |
| ≤1 | 3 | 49 | 1 | 1 | |
| ≥2 | 62 | 288 | 3.51(1.06,11.6) | 3.07(0.67,14.08) | 0.149 |
| **Knowledge on danger signs of pregnancy** | | | | | |
| Not Knowledgeable | 14 | 243 | **1** | **1** | |
| Knowledgeable | 51 | 145 | **9.4(4.9,17.81)** | **7.88(3.72,16.69)** | **0.000** |
| **Awareness on EDD** | | | | | |
| Not aware | 16 | 122 | 1 | 1 | |
| Aware | 49 | 225 | 1.52(.83,2.8) | 1.16(0.51,2.67) | 0.718 |
| **Decision on mother's health** | | | | | |
| mother | 10 | 79 | 1 | 1 | 0.027 |
| husband | 12 | 110 | .862(.35, 2.09) | 1.09(0.37,3.25) | 0.879 |
| Jointly | 43 | 148 | 2.29(1.09,4.8) | **2.76(1.08,7.05)** | **0.034** |
| **Getting people for house holed activities** | | | | | |
| Possible | 30 | 83 | 2.65(1.52,4.5) | **2.59(1.21,5.52)** | **0.014** |
| Not possible | 35 | 254 | 1 | 1 | |
| **Getting chilled caregiver** | | | | | |
| Possible | 25 | 89 | 1.74(.99, 3.03) | 0.74(0.31,1.77) | 0.499 |
| Not possible | 40 | 248 | 1 | 1 | |
| **Acceptable waiting time at MWH(2-4wks)** | | | | | |
| Acceptable | 42 | 107 | **3.92(2.24,6.85)** | **3.15(1.54,6.43)** | **0.002** |
| Not Acceptable | 23 | 230 | **1** | **1** | |

Although distance did not show an association with the MWHS utilization in this study, evidence revealed that distance from the nearby health facility is one determinant factor for MWHS utilization [22]. The other explanation for the higher proportion might be the time gap. Nowadays, maternal health is a global priority area, and special focus might be given to increasing MWHS utilization.

The present study indicates that maternal age is one significant factor for MWHs utilization. Thus, mothers in the age category of 26–30 were 76% less likely to utilize MWHS than those women aged above 36 years old. This might be due to the fact that aged mothers might have matured children, which may have overtaken the overall household activities. In addition, those older mothers may have had past bad obstetric experiences and be worried about a recurrence of history, thereby utilizing MWHS. This conclusion is supported by other studies in Ethiopia [17, 18] and Zambia [29]. Also, older women have a greater chance of visiting health institutions and may get contacted with healthcare providers, thereby getting adequate information about maternity health services, including MWHS. Moreover, older women may have higher decision-making autonomy in the household on maternal and children health-related issues [30], so they will decide to utilize every maternal health service, including MWHS.

This study also revealed that mothers attending the primary level of education were 9.05 times more likely to utilize MWHS compared with their counterparts. This finding is supported -by a study done in the Hadya zone, southern Ethiopia, which showed that educated mothers were more likely to intend to utilize MWHs than non-educated ones [21]. This might be due to the reality that education increases the likelihood of risk perception, level of understanding, and easy acceptance of health-related information and advice. As a result, educated women will take care of their health and their pregnancy.

Another relevant finding of the current study is that women having adequate knowledge of pregnancy danger signs were 7.88 times more likely to utilize MWHS as compared to women who had no adequate knowledge. This might be justified as mothers having a better knowledge of the danger signs of pregnancy will have a high perception of the occurrence of danger signs and will consider utilizing MWHS as one preservative method. In this regard, health care providers at ANC services and health extension workers during home-to-home visits should emphasize educating and counseling about the danger signs of pregnancy for pregnant women.

The present study affirmed that women who had shared decisions with their husbands regarding their health were 2.76 times more likely to utilize MWHs as compared to their counterparts. The reason might be that women who have their husbands involved in their health and who receive support on different household duties will use MWHS freely. Previous studies support this finding in which women who had experienced disagreements or challenges from their husbands or other family members were not able to utilize MWHS [9].

Moreover, this study found that mothers who had gotten people to cover household activities were 2.59 times more likely to use MWHS than women who had not gotten people to cover household activities. This is because women who had an additional person replace their work at home may have free time and can easily access health care services [31].

Lastly, this study revealed that mothers who had accepted the specified duration of waiting time were 3.15 times more likely to utilize MWHS than those mothers who hadn't accepted the specified waiting time duration. According to evidence, pregnant women preferred shorter lengths of stay (less than 14 days) at MWHS [32]. This is due to the fact that the women's concerns might arise from a lack of caregivers for their children or household chores while waiting for a long time in the MWH. Admitting pregnant women to MWH far from their expected date of delivery might be challenging to fulfill basic facilities, and their families face difficulties.

## Strength and limitations of the study

This study has its own strengths and limitations. We believe that this study will have good input on the existing gap regarding MWHS utilization and will help reduce maternal and

perinatal mortalities. However, the study has some limitations in which the readers need to consider during interpretation. First, the cross-sectional nature of the study may not clearly show the effect of the suggested predictors on MWHS utilization. Second, the study tried to illustrate quantitative factors, but behavioral, social, and cultural factors which by nature need qualitative research were not assessed. In this regard, further qualitative researches might be needed.

## Conclusion

The magnitude of maternity waiting home-service utilization was low in this study. Primary level of education, accepting length of stay, knowing danger signs of pregnancy and a mother who decided on her health jointly with her husband were all positively associated with MWH-service utilization, whereas maternal age 26-30-year-old was a negative associated with MWH-service utilization. Thus, health education communication regarding danger signs of pregnancy, empowering the woman's decision making, educating the adults at list the primary level of education, and shortening the length of stay at MWH may enhance MWHS utilization.

## Supporting information

**S1 File. English and Amharic versions of the questionnaire.**
(DOCX)

**S2 File. SPSS dataset.**
(SAV)

## Acknowledgments

We would like to thank all data collectors and study participants. We are also glad to thank Dabat district administrative health office for writing a permission letter.

## Author Contributions

**Conceptualization:** Mulugeta Melese Shiferaw.

**Data curation:** Mulugeta Melese Shiferaw, Agumas Eskezia Tiguh, Azmeraw Ambachew Kebede, Birhan Tsegaw Taye.

**Formal analysis:** Mulugeta Melese Shiferaw, Agumas Eskezia Tiguh, Azmeraw Ambachew Kebede, Birhan Tsegaw Taye.

**Funding acquisition:** Mulugeta Melese Shiferaw, Agumas Eskezia Tiguh, Azmeraw Ambachew Kebede, Birhan Tsegaw Taye.

**Investigation:** Mulugeta Melese Shiferaw, Agumas Eskezia Tiguh, Azmeraw Ambachew Kebede, Birhan Tsegaw Taye.

**Methodology:** Mulugeta Melese Shiferaw, Agumas Eskezia Tiguh, Azmeraw Ambachew Kebede, Birhan Tsegaw Taye.

**Project administration:** Mulugeta Melese Shiferaw, Agumas Eskezia Tiguh, Azmeraw Ambachew Kebede, Birhan Tsegaw Taye.

**Resources:** Mulugeta Melese Shiferaw, Agumas Eskezia Tiguh, Azmeraw Ambachew Kebede, Birhan Tsegaw Taye.

**Software:** Mulugeta Melese Shiferaw, Agumas Eskezia Tiguh, Azmeraw Ambachew Kebede, Birhan Tsegaw Taye.

**Supervision:** Mulugeta Melese Shiferaw, Agumas Eskezia Tiguh, Azmeraw Ambachew Kebede, Birhan Tsegaw Taye.

**Validation:** Mulugeta Melese Shiferaw, Agumas Eskezia Tiguh, Azmeraw Ambachew Kebede, Birhan Tsegaw Taye.

**Visualization:** Mulugeta Melese Shiferaw, Agumas Eskezia Tiguh, Azmeraw Ambachew Kebede, Birhan Tsegaw Taye.

**Writing – original draft:** Mulugeta Melese Shiferaw.

**Writing – review & editing:** Mulugeta Melese Shiferaw, Agumas Eskezia Tiguh, Azmeraw Ambachew Kebede, Birhan Tsegaw Taye.

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
