## [Decision Letter · Decision Letter 0]

19 Jan 2022

PONE-D-21-25855Utilization of maternal waiting home and associated factors among women who gave birth in the last one year, Dabat district, Northwest EthiopiaPLOS ONE

Dear Dr. Tiguh,

Thank you for submitting your manuscript to PLOS ONE. After careful consideration, we feel that it has merit but does not fully meet PLOS ONE’s publication criteria as it currently stands. Therefore, we invite you to submit a revised version of the manuscript that addresses the points raised during the review process.

ACADEMIC EDITOR: General commentsThe topic is quite relevant towards reducing the burden of maternal and perinatal morbidity and mortality in the region. The manuscript however has a lot of grammatical errors which needs to be corrected.BackgroundThe justification for the study has not been clearly written. Authors needs to clearly justify why this study is relevant in the region, especially when there are several studies relating to utilization of maternity waiting homes in Ethiopia.Materials and methodsThe methodology is appropriate, but authors may need to justify the selection of variables with p-value of less than 0.2 instead of 0.05 for inclusion into multivariate logistic regression analysis.ResultThe result section needs to be revised with absolute values written before the percentage in brackets.DiscussionThe authors have not discussed the findings with reference to the main objectives of the study rather there are repetitions of the results in several aspects of the discussion. The grammatical errors also made the flow cumbersome. The strengths and limitations of the study should be clearly highlighted.Both reviwers have also raised the challenge of numerous gramatical errors in themanuscript which needs to be corrected.I recommend use of an English Language editor if possible.Please also ensure that the references are as per journal recommendations(Vancouver referencing style)as many of the quoted refrences do not conform with this style.Other comments are as shown below Please submit your revised manuscript by Mar 05 2022 11:59PM. If you will need more time than this to complete your revisions, please reply to this message or contact the journal office at plosone@plos.org. Please include the following items when submitting your revised manuscript:A rebuttal letter that responds to each point raised by the academic editor and reviewer(s). You should upload this letter as a separate file labeled 'Response to Reviewers'.A marked-up copy of your manuscript that highlights changes made to the original version. You should upload this as a separate file labeled 'Revised Manuscript with Track Changes'.An unmarked version of your revised paper without tracked changes. You should upload this as a separate file labeled 'Manuscript'.

We look forward to receiving your revised manuscript.

Kind regards,

Godwin Otuodichinma Akaba, MBBS,MSc,MPH,FWACS

Academic Editor

PLOS ONE

https://journals.plos.org/plosone/s/file?id=ba62/PLOSOne_formatting_sample_title_authors_affiliations.pdf”

A clean copy of the edited manuscript (uploaded as the new *manuscript* file).

3. Please amend your current ethics statement to address the following concern: Please explain i) why written consent was not obtained, ii) how you documented participant consent, and iii) whether the ethics committees/IRB approved this consent procedure.

“university of Gondar, Ethiopia”

6. Thank you for stating the following in the Funding Section of your manuscript:

“This study was sponsored by the University of Gondar.”

“university of Gondar, Ethiopia”

7. Your ethics statement should only appear in the Methods section of your manuscript. If your ethics statement is written in any section besides the Methods, please delete it from any other section.

Abstract

Background

Item: page 1,lies 3-4:To alleviate this problem, maternity waiting homes are a get way for women to deliver at health facility thereby help in reducing the alarming

Comment: Change to: gateway for women to deliver at health facility thereby helping towards the reduction of the alarming……

Item: Page 1, line 6: However, there is a paucity of evidence in this regard in the study area

Comment: The author has not described the evidence that is lacking which has heralded the study. Please rephrase to: However, there is a paucity of evidence regarding the utilization of these facilities by pregnant women in the study area.

Introduction

Item:Page 10,lies 7-9: Ethiopia is also one of the SSA countries with low maternal health services utilization and high materiality mortality. Thus, the facility birth rates are as low as 28 % and about 1400 maternal deaths were occurred annually (2)

Comment: Paraphrase to: Ethiopia is also one of the SSA countries with low maternal health services utilization and high maternal mortality ratio. Thus, the facility birth rates are as low as 28 % and about 1400 maternal deaths occur annually (2)

Item;page 11,lies 7-11: In addition, it urges women to use maternal healthcare service like skilled birth attendants and other comprehensive emergency obstetrics care thereby reducing negative pregnancy outcomes apart from maternal mortality and timely utilization of MWHs effectively negative pregnancy outcomes(12,14,15). Moreover, MWHs can decrease the gap between urban-rural maternal health service utilization (5).

Comment: delete the last part of the paragraph ad paraphrase to:

In addition, it urges women to use maternal healthcare service like skilled birth attendants and other comprehensive emergency obstetrics care thereby reducing negative pregnancy outcomes (12,14,15). Moreover, MWHs can decrease the gap between urban-rural maternal health service utilization (5).

Item: page 11,lines 12-14: Some of the published studies focus merely on the physical establishment of MWHs at health institutions rather than the utilization , some other studies focus on the intention to use rather than actual utilization of MWHs

Comment: Please provide the references for the above statements.

Item:Page 12:Independent variables: Socio-demographic characteristics; Age of mother, religion, marital status, Mother's Occupation, Mother's Education, Partner's Education. Distances to the health facilities, Transportation access to the health facilities, affordability of transport cost, way of transportation

Comment: Start other words apart from the first in small letters.

Socio-demographic characteristics; Age of mother, religion, marital status, occupation, educational status, partner's educational status. distance to the health facility, transportation access to the health facility, affordability of transport cost, type of transportation

Item: Reproductive health and obstetrics related; Decision power of mother on own health, number of live birth, History of stillbirth, Birthplace of the last child, No ANC visits of the last pregnancy, planed or un planed pregnancy, place of ANC visit, information on Birth preparedness plan, Knowledge of danger sign during pregnancy, Awareness of expected date of delivery

Comment: Start other words apart from the first in small letters(see previous comment above).

Operational definitions

Item: Knowledge on danger sign of pregnancy: a woman who list out three and more danger signs of pregnancy (Vaginal bleeding, gush of fluid per vagina, severe abdominal pain, high grade fever, fainting, decreased fetal movement, blurred vision, severe headache, edema or body swelling) considered as knowledgeable (19).

Comments: Why should a woman who mentions three out of nine listed danger signs be termed as knowledgeable. In all forms of assessment 33% cannot be termed as knowledgeable.

Item: Accepted length of stay: women’s perceived to the length of two-four weeks’ duration is optimal time.

Comment: This statement is ot clear. Please paraphrase

Item:page 13,lines 17-18: To maximize the reliability and validity of the variables in the study, special attention was given gave to the construction of the questionnaire,

Comment: To maximize the reliability and validity of the variables in the study, special attention was given to the construction of the questionnaire,………..

Method

Why was the p value for variables to be included in the multivariate logistic analysis set at p <0.2 and not 0.05? It would have been nice to see how many variables that would have been favoured if it was 0.05.The reduction to 0.2 implied that almost all the variables were part of the multivariable logistic regression analysis.

Results

Item: Page 14, lines 21-23: and half (51%) of their husbands has no formal education. More than one fourth (23.6%) of mothers two hour and more time taken to reach the nearest health institution….

Comment: Authors should write the absolute figure before the percentage in bracket. In the above statement half is not 51% but 50%. Similarly, more than one fourth implies that it is more than 25%.

All aspects of the results should be written in the suggested format(y%) instead of the current style of approximate statements before the percentage.

Discussion

This section has a lot of grammatical errors that need to be extensively revised. I have highlighted some, but authors should please go through the remaining paragraphs in the manuscript to correct these errors.

Item: page 16,line: The discrepancies might appeared from the differences in study setting,…..

Comment: Paraphrase to: The discrepancies may be due to the differences in study setting,…….

Item: Page 16,lines 17-19: was community-based For instance, the study in Jimma district includes all women who were gave birth in health facilities, so the chances to obtain MWH- users might be high among those women

Comment: Paraphrase to : was community-based. For instance, the study in Jimma district included all women who gave birth in health facilities, so the chances to obtain MWH- users might be high among those women.

Item: Page 16,lines 21-22: It is believed that once they admitted to the MWHs, they had a great chance of giving birth at the health facilities.

Comment: Paraphrase to: It is believed that admission into the MWHs increases the chances of the women giving birth at the health facilities.

Strengths and limitations

These are missing in this manuscript and should be added in the revised manuscript.

Table 1: Mothers occupation

Comment: The total percent for the above variable in the table is 99.9% instead of 100%.

Table 1: Husband/father of child/ Level of education

Comment: The total percent for the above variable in the table is 99.9% instead of 100%.

Time takin to the nearest Health center

Comment: The total percent for the above variable in the table is 99.9% instead of 100%.

These challenges may have been resolved if the approximation was to two decimal points.

Table 4: Factor associated with maternal waiting home utilization, Dabat district, north west Ethiopia,2019

Comment: Please include a column for the p-values

Table 4:

Gaining people for house holed activities: Getting people for household activities

Gating chilled care giver: Getting childcare giver

References

Authors have not adopted the Vancouver referencing style in this manuscript. The reference section should be extensively revised to conform with the recommended refencing style for the journal.

Figure 1: know the plase of mwh

Comment: Patient knows the location of mwh

Figure 2:Is not relevant and can be deleted.

Reviewers' comments:

Reviewer's Responses to Questions

**Comments to the Author**

1. Is the manuscript technically sound, and do the data support the conclusions?

Reviewer #1: Yes

Reviewer #2: Partly

2. Has the statistical analysis been performed appropriately and rigorously? 

Reviewer #1: Yes

Reviewer #2: No

3. Have the authors made all data underlying the findings in their manuscript fully available?

Reviewer #1: Yes

Reviewer #2: Yes

4. Is the manuscript presented in an intelligible fashion and written in standard English?

Reviewer #1: Yes

Reviewer #2: No

5. Review Comments to the Author

Reviewer #1: This is an interesting study aimed at assessing the utilization of maternity waiting home services

and factors associated among mothers who gave birth in the last one year prior to the

study period in Dabat district.

The manuscript needs a language editor.

TITLE

This is ok.

ABSTRACT

What are the inclusion and exclusion criteria? what are the outcome measures?

INTRODUCTION

This is well written. However, the authors should beef up the justifications for the study.

METHODS

There are so many typographical errors.

Eg:' Sample size determination and sapling procedure '. It is supposed to be sampling.

RESULTS

The authors should begin with stating how many were assessed for eligibility and how many were excluded with reasons.

The authors should include a flowchart to illustrate the participants flow.

DISCUSSION

What are the strengths and limitations

CONCLUSION . This is OK.

References

This is ok.

Reviewer #2: Thank you authors for the great work. As this was a population based study the findings could be generalizable. However, it is not clear how household or individuals were selected. Did you do cluster sampling? in stages or what? Please in a sentence address your sampling procedure clearly.

There a countless grammatical/spelling errors in both heading and texts. Sampling is typed as Sanpling; population as pollution, etc.

The discussion has not been robust. Authors should restate their main objective in this discussion section and attempt addressing same.

There is no clear cut strength and limitation of this study.

6. PLOS authors have the option to publish the peer review history of their article (what does this mean?). If published, this will include your full peer review and any attached files.

Reviewer #1: **Yes: **George Uchenna Eleje

Reviewer #2: No

---

## [Decision Letter · Decision Letter 1]

30 May 2022

PONE-D-21-25855R1Utilization of maternal waiting home and associated factors among women who gave birth in the last one year, Dabat district, Northwest EthiopiaPLOS ONE

Dear Dr.Tiguh,

Thank you for submitting your manuscript to PLOS ONE. After careful consideration, we feel that it has merit but does not fully meet PLOS ONE’s publication criteria as it currently stands. Therefore, we invite you to submit a revised version of the manuscript that addresses the points raised during the review process.

ACADEMIC EDITOR: 

The manuscript has extensively been revised but there are still some typographical and grammatical mistakes which needs to be corrected. Authors should carefully go through the entire manuscript to ensure that these have all been corrected.

Additionally, the following observed mistakes should be corrected during the revision of this manuscript

Table 4:P value

Comment: Include the zero before the decimal points

Discussion

Page 17, lines 22-24: Moreover, older women have higher decision-making autonomy in the household and maternal and children health [30]

Change to: Moreover, older women have higher decision-making autonomy in the household on maternal and children health related issues [30]

Page 17, line 26: This study also revealed that mothers attending the primary level of education was 9.05 time

Change to: This study also revealed that mothers attending the primary level of education were 9.05 time

Page 17, lines 27,28,29: This finding is supported by a study done in Hadya zone, southern Ethiopia, which showed that educated mothers were more likely to intended to utilize MWHs than non-educated ones [21]

Change to: This finding is supported by a study done in Hadya zone, southern Ethiopia, which showed that educated mothers were more likely to intend to utilize MWHs than non-educated ones [21]

Page 17, last line: As a result, women will take care of their health and their pregnancy.

Change to: As a result, educated women will take care of their health and their pregnancy.

Page 18: Previous studies support this finding in which women who had experienced of disagreements or challenges from their husbands or other family members were not able to utilize MWHs

Change to” Previous studies support this finding in which women who had experienced disagreements or challenges from their husbands or other family members were not able to utilize MWHs

We look forward to receiving your revised manuscript.

Kind regards,

Godwin Otuodichinma Akaba, MBBS,MSc,MPH,FWACS

Academic Editor

PLOS ONE

**Journal Requirements:**

**Additional Editor Comments: **

The manuscript has extensively been revised but there are still some typographical and grammatical mistakes which needs to be corrected. Authors should carefully go through the entire manuscript to ensure that these have all been corrected.

Additionally, the following observed mistakes should be corrected during the revision of this manuscript

Table 4:P value

Comment: Include the zero before the decimal points

Discussion

Page 17, lines 22-24: Moreover, older women have higher decision-making autonomy in the household and maternal and children health [30]

Change to: Moreover, older women have higher decision-making autonomy in the household on maternal and children health related issues [30]

Page 17, line 26: This study also revealed that mothers attending the primary level of education was 9.05 time

Change to: This study also revealed that mothers attending the primary level of education were 9.05 time

Page 17, lines 27,28,29: This finding is supported by a study done in Hadya zone, southern Ethiopia, which showed that educated mothers were more likely to intended to utilize MWHs than non-educated ones [21]

Change to: This finding is supported by a study done in Hadya zone, southern Ethiopia, which showed that educated mothers were more likely to intend to utilize MWHs than non-educated ones [21]

Page 17, last line: As a result, women will take care of their health and their pregnancy.

Change to: As a result, educated women will take care of their health and their pregnancy.

Page 18: Previous studies support this finding in which women who had experienced of disagreements or challenges from their husbands or other family members were not able to utilize MWHs

Change to” Previous studies support this finding in which women who had experienced disagreements or challenges from their husbands or other family members were not able to utilize MWHs

Reviewers' comments:

Reviewer's Responses to Questions

**Comments to the Author**

1. If the authors have adequately addressed your comments raised in a previous round of review and you feel that this manuscript is now acceptable for publication, you may indicate that here to bypass the “Comments to the Author” section, enter your conflict of interest statement in the “Confidential to Editor” section, and submit your "Accept" recommendation.

Reviewer #1: All comments have been addressed

Reviewer #2: All comments have been addressed

2. Is the manuscript technically sound, and do the data support the conclusions?

Reviewer #1: Yes

Reviewer #2: Partly

3. Has the statistical analysis been performed appropriately and rigorously? 

Reviewer #1: Yes

Reviewer #2: No

4. Have the authors made all data underlying the findings in their manuscript fully available?

Reviewer #1: Yes

Reviewer #2: No

5. Is the manuscript presented in an intelligible fashion and written in standard English?

Reviewer #1: No

Reviewer #2: No

6. Review Comments to the Author

Reviewer #1: The authors have responded adequately. They have corrected all the typographical errors. There is need to address all issues raised

Reviewer #2: The study is interesting and present a great strategy for improvement of maternal and newborn health. There is however some concerns. The study design has not been stated and rationale for sample size has not been described. The discussion has not been robust as literature reviews have been sparse and uncoordinated. Authors have also not stated strength and limitation of their study.

Gramma errors such as pollution instead of population, sapling instead of sampling, etc need to be corrected.

7. PLOS authors have the option to publish the peer review history of their article (what does this mean?). If published, this will include your full peer review and any attached files.

Reviewer #1: **Yes: **George Eleje

Reviewer #2: **Yes: **Emmanuel Ugwa

---

## [Editor Report · Decision Letter 2]

24 Jun 2022

Utilization of maternal waiting home and associated factors among women who gave birth in the last one year, Dabat district, Northwest Ethiopia

PONE-D-21-25855R2

Dear Dr Tiguh,

We’re pleased to inform you that your manuscript has been judged scientifically suitable for publication and will be formally accepted for publication once it meets all outstanding technical requirements.

Kind regards,

Godwin Otuodichinma Akaba, MBBS,MSc,MPH,FWACS

Academic Editor

PLOS ONE
---

## [Editor Report · Acceptance letter]

29 Jun 2022

PONE-D-21-25855R2 

Utilization of maternal waiting home and associated factors among women who gave birth in the last one year, Dabat district, Northwest Ethiopia 

Dear Dr. Tiguh:

I'm pleased to inform you that your manuscript has been deemed suitable for publication in PLOS ONE. Congratulations! Your manuscript is now with our production department. 

Kind regards, 

on behalf of

Dr. Godwin Otuodichinma Akaba 

Academic Editor

PLOS ONE